# Attitudes and Barriers of Polish Women towards Breastfeeding—Descriptive Cross-Sectional On-Line Survey

**DOI:** 10.3390/healthcare12171744

**Published:** 2024-09-02

**Authors:** Agnieszka Kolmaga, Katarzyna Dems-Rudnicka, Anna Garus-Pakowska

**Affiliations:** 1Department of Nutrition and Epidemiology, Medical University of Lodz, 90-752 Lodz, Poland; agnieszka.kolmaga@umed.lodz.pl; 2Centre of Mathematics and Physics, Lodz University of Technology, 90-924 Lodz, Poland; katarzyna.dems@p.lodz.pl

**Keywords:** breastfeeding, mothers, decision, attitudes, barriers, Poland

## Abstract

Background: Breastfeeding is the gold standard in infant nutrition. Successful breastfeeding depends on many factors, including the help of medical personnel in teaching breastfeeding, the need for professional work, and breastfeeding-friendly places in public spaces. The main goal was to identify various barriers among mothers to breastfeeding. Methods: This study used a quantitative descriptive research design. We recruited 419 mothers aged at least 18 years old through social media. Results were analysed using Pearson’s chi-squared and Fisher’s tests of independence for pairs of dependent and independent variables. Results: Most often, women gave birth at the age of 25–30, had one or two children, and attended higher education. Almost half of them lived in a large city and gave birth to a child by caesarean section. A total of 83.1% of mothers planned to breastfeed, but not all of them were able to do so for various reasons. One-third of them felt sorry for themselves that they had to change their feeding method. The majority of mothers did not receive sufficient help in the hospital in terms of learning how to breastfeed (61%), did not use the help of a lactation consultant (67%), and answered that there was no lactation consultant in their place of residence (65%). Only 43.2% of mothers returned to work without ceasing breastfeeding. A total of 42% of mothers experienced feelings of embarrassment when breastfeeding in a public place. The most frequently indicated barrier to breastfeeding in a public place was the lack of a suitable location where a woman would feel comfortable, calm, and intimate. Conclusion: There are various barriers towards breastfeeding: too few lactation consultants, problems with breastfeeding when the mother wants to return to work, and unfriendly places for breastfeeding in public spaces. Efforts must be made to support mothers in breastfeeding.

## 1. Introduction

Public health organisations stress that breastfeeding is a key objective for achieving a healthy population now and in the long term [1,2]. The benefits of natural feeding are undeniable and well established for both mother and child, as emphasised by the medical community, numerous scientific organisations, and progressive social media [3,4,5,6,7,8], which also translates into far-reaching social and environmental benefits and generates significant economic savings for the country [9,10].

Breastfeeding is an essential element of optimal nutrition of newborns and infants up to 6 months old, as it affects proper growth and psycho-physical development. Maternal milk is a mother’s most precious gift to her child due to the health-promoting properties derived from its composition [11]. Breast milk is the most preferred—natural, complete, and safe—food during this period of life, containing all the nutrients and antibodies necessary for the proper growth and development of the baby [4,5,7,8,10,11,12].

According to the World Health Organization [WHO], exclusive breastfeeding (EBF) is widely recognised as the gold standard for feeding a baby up to 6 months of age [6]. In the second half of the first year of life, it can provide more than half of the necessary nutrients and can be continued depending on the child’s needs until 2 years of age [13,14], gradually incorporating complementary foods into the toddler’s diet [12,15]. The first 1000 days of life, including pregnancy and the child’s first 2 years, have been identified as a critical period that can have a positive or negative impact on human development throughout life [10].

Breastfeeding has many advantages and is extremely important for both mother and baby. This food has a unique composition that adapts to the needs of the child, including a wide group of bioactive compounds, including proteins/peptides, oligosaccharides, hormones, nucleotides, vitamins, minerals, and innate immune factors [13,14,16], in addition to being the strongest predictor of gut microbiota composition in the first months of life [14]. The benefits associated with breastfeeding (BF), therefore, include protection against pathogens, complete nutrition, enhanced child and immune development, promotion of intestinal colonisation, and reduced incidence of gastrointestinal, nervous, respiratory, and other diseases [11,16]. Breastfeeding is also an important issue for women’s health; long-term breastfeeding is associated with a reduced risk of inflammatory diseases, allows you to lose weight faster and avoid metabolic complications associated with obesity, and reduces the risk of cardiovascular diseases and certain cancers [e.g., breast cancer, ovarian cancer]. Moreover, breastfeeding reduces the risk of postpartum depression and promotes better sleep quality, especially in early motherhood. The direct benefits of breastfeeding are also a shorter postpartum bleeding period, faster uterine involution, and lactational infertility. In addition, it allows a strong bond to be formed between mother and child, and, above all, it is cheap, easily accessible, and convenient [5,10,11,13,14,16,17].

Despite the well-known benefits of breastfeeding, current rates of exclusive breastfeeding are not optimistic [11,12,18,19]. Low rates and early cessation of breastfeeding have serious negative health consequences for infants, young children, and women [20].

According to Cordero et al. [10], there has been a slow global increase in exclusive breastfeeding [EBF] in different regions, but breastfeeding practices worldwide are not optimal. The vast majority of countries are still far from achieving the WHO’s recently updated EBF target, which is that all countries should reach at least 70% EBF prevalence among infants under 6 months of age by 2030 [10]. Among the world regions, the WHO European region has the lowest rate of exclusive breastfeeding at 6 months of age—about 25. In studies in 11 European countries, 56–98% of infants received breast milk immediately after birth, but only 38–71% of infants at 6 months of age received female formula, and only 13–39% of European infants were exclusively breastfed [20].

Data from 2014 to 2020 show that the percentage of exclusive breastfeeding of infants under 6 months of age is only 44% for the global population [21]. Breastfeeding at the beginning of lactation is quite high in Poland (97–99.4%) [5,22,23,24] and in Australia (95.9%) [1]. A total of 93–78% of infants are breastfed in Sweden after birth [19] and 81% in the USA [25]. This is a further percentage of women who naturally breastfeed about about 6 months of life, which significantly declines. In Poland, women who continue exclusive breastfeeding up to 6 months are of concern (from 4 to 22.4%) [23]. In studies by Morns et al. in Australia, 66.0% of women exclusively breastfed their child up to 4 months [1]. In the USA, only about 25.5% of women exclusively breastfed their child up to 6 months postpartum [25], and in Sweden, the percentage of exclusive breastfeeding fell to about 15% [19].

Mothers’ uncertainty regarding whether their child’s nutritional needs are met solely by breast milk is identified as the primary rationale for mothers to introduce mixed feeding, as opposed to exclusively breastfeeding, particularly in infants under the age of three months [15,19]. However, it appears that this phenomenon is more dependent on the subjective assessment of the mother herself than on objective issues related to lactation. The practice of feeding infants a combination of breast milk and formula is becoming increasingly prevalent worldwide [23,26,27]. It is frequently reported that mothers use a bottle at night to facilitate peaceful sleep for themselves and their infants. In addition, the decision to practice mixed feeding was based on the perceived benefits of combining breastfeeding with milk formula: “for example, the health aspects of BF, with the convenient aspects of bottle feeding…” [15,27].

Protecting, promoting, and supporting breastfeeding should, therefore, be a public health priority in all countries, especially those with low rates [20,28]. Breastfeeding is a shared social responsibility where we should all contribute to creating an environment that promotes, protects, and supports breastfeeding [1,10,28].

To improve infant and maternal health care, including breastfeeding rates, the overriding objective is to take all appropriate measures to ensure successful breastfeeding, including early identification of potential breastfeeding problems and knowledge about the attitudes and awareness of women regarding the benefits of natural feeding undertaken [24].

We hope that the study will provide new, up-to-date data and allow us to better/effectively focus on promoting natural breastfeeding in Poland, taking into account the barriers to its adoption and attitudes that affect breastfeeding.

The main goal was to identify various barriers among mothers preventing exclusive breastfeeding.

The specific aim was to answer the following questions:How women planned to feed their baby and how they actually fed;What barriers might be preventing breastfeeding.

## 2. Materials and Methods

### 2.1. Study Design

This study employed a quantitative descriptive research design to achieve the study’s objectives. The participants were recruited during the autumn of 2022 among users of Polish portals and social media for mothers. Their selection was based on keywords entered in the search engine: “mothers”, “infants”, and “feeding infants”. The groups designated as “Moms of Children” and “Mothers” were identified. The two groups had approximately 18.5 thousand members from across Poland. The second criterion was the reach of the influence, with the intention of forming a cross-sectional group comprising individuals from a range of geographical locations. Thus, we excluded all so-called “private” and local groups. To ensure a representative sample, groups with a specific focus on breastfeeding were excluded, as women with a strong interest in this topic are likely to be active in such groups. Additionally, the opinions of mothers who planned to feed their children with formula milk were included in the study.

In Poland, the number of births has been in decline on an annual basis, with 355 thousand live births occurring between 2020 and 2022 (data from the Central Statistical Office: stat.gov.pl).

The fertility rates were as follows: 1.387 in 2020, 1.33 in 2021, and 1.26 in 2022. It is noteworthy that the coefficient ensuring simple generational replacement is assumed to be 2.1–2.15. The demographic profile of the study participants was comparable to that of the general female population in Poland and aligned with the metric variables observed in other studies [29].

### 2.2. Sample and Data Collection

The study was conducted with the participation of women aged 18 years and above. The link to the survey was shared via social media in a nationwide group dedicated to the topic of motherhood. In addition to the criterion of age, participation in the study was contingent upon having at least one child born within a maximum of five years preceding the study.

Information regarding the study was disseminated via Google Forms, along with a link to the survey. Furthermore, the survey commenced with an introductory section, which provided respondents with information regarding the study’s objectives, instructions on how to complete the questionnaire, details on the voluntary nature of participation, and the option to discontinue the questionnaire at any time (Appendix A).

A total of 433 women completed the survey. Of these, 419 fully completed questionnaires that met the study criteria were qualified for inclusion in the study.

Given the nature of the study, approval from a bioethics committee was not required. The questionnaire was conducted in accordance with the principles of ethics [30].

### 2.3. Research Tool

The questionnaire was developed by the authors based on an analysis of the existing literature and their collective expertise (epidemiologist and dietitian). It was subsequently refined through the conducting of a study involving a smaller group of mothers by a public health student as part of her diploma thesis. This pilot study made it possible to check whether the questions were understandable.

The survey questionnaire was comprised of three distinct sections, each containing both single-choice and multiple-choice questions time (Appendix A).

The initial section of the questionnaire pertained to the characteristics of the participants. The variables subjected to analysis were the socio-demographic characteristics of the respondents, including age, level of education, place of residence, number of children, and age at which the woman gave birth to her first child. Due to the size of the groups, we aggregated the answers for rural and city <100,000 inhabitants groups for the calculations. Additionally, the questionnaire inquired about the method used to terminate the pregnancy.

The second section of the questionnaire addressed opinions regarding breastfeeding. For example, whether formula milk is as valuable as breast milk, and what are the benefits of breastfeeding (for both the baby and its mother). In addition, respondents were asked to assess their knowledge of infant nutrition on a five-point Likert scale. When answering this question, respondents determined their level of knowledge (or lack of knowledge) on a symmetrical scale, where point 1 meant “I have no knowledge” and point 5 meant “very good knowledge”.

The third section comprised questions pertaining to the factors that inform decisions regarding breastfeeding, encompassing both potential impediments and advantages. The identified barriers were the comfort of breastfeeding in public places, the ability to return to work while breastfeeding, and the assistance of a lactation consultant. We also asked whether having a caesarean section was a barrier to breastfeeding.

### 2.4. Statistical Analysis

The data set under consideration comprised dependent variables (answers) and 419 independent variables (respondents).

Pearson’s chi-squared (X^2^) and, on occasion, Fisher’s tests of independence for pairs of dependent and independent variables were employed to investigate the potential stochastic relationship between variables describing the general population. Furthermore, to assess the precision of the estimated proportions, we employed the Jeffreys confidence intervals [31]. The statistical computations were conducted using R ver. 4.4.0 [32]. The level of statistical significance was set at 0.05.

## 3. Results

### 3.1. Characteristics of the Study Group

The study group comprised 419 women aged 18 or above. Table 1 presents a summary of the socio-demographic characteristics of the study group. The majority of women in the study group had given birth between the ages of 25 and 30 and had one or two children. The majority of the women had obtained a higher education, a finding that is consistent with the results of other Polish studies (29). Approximately 47% of the respondents resided in a metropolitan area with a population exceeding 100,000. In nearly 47% of cases, the pregnancy was terminated by caesarean section, which corresponds to the national average.

The majority of respondents (77.3%, *n* = 324) indicated that they possessed sufficient knowledge about breastfeeding. Furthermore, the women surveyed indicated a relatively high level of knowledge regarding infant nutrition. A total of 51.8% and 31.9% of respondents, respectively, rated their knowledge as excellent (level 4 and 5 on the Likert scale). No respondents selected the first option on the Likert scale (“I have no knowledge”).

### 3.2. Women’s Attitudes towards Breastfeeding

The overwhelming majority of women (83.1%) indicated that they intended to breastfeed their child (Table 2). There were notable differences between the planned and actual methods of breastfeeding (X^2^ = 103.01, df = 4, *p*-value < 2.2 × 10^−16^).

The future mother’s preferred method of feeding her child exhibited a statistically significant correlation with her level of education (X^2^ = 13.291, df = 4, *p*-value = 0.009939) and the number of children (X^2^ = 14.698, df = 6, *p*-value = 0.02274). These mothers were more likely to select a mixed feeding approach. The preference for a mixed feeding plan was also more prevalent among women aged 30 < 40 (39.6%) and those residing in urban areas (52%). However, this trend did not reach statistical significance.

It should be noted that not all mothers chose to breastfeed and that there were a number of reasons why this was the case. Of the 348 women who expressed a desire to breastfeed, 255 initiated lactation, while 14 opted for formula feeding. The remaining participants opted to employ a combination of both feeding methods (*n* = 79). However, the opposite also occurred: of the 23 women who had indicated their intention to feed with formula milk, five opted to breastfeed, while eight switched to mixed feeding.

At the same time, one in three mothers who were unable to breastfeed (30.7%) reported feelings of resentment towards themselves for changing their feeding method. This was particularly prevalent among women who became mothers at a later age (X^2^ = 13.112, df = 6, *p*-value = 0.04129).

A total of 21% of respondents (*n* = 89) indicated that they considered formula milk to be as valuable as mother’s milk, which was not significantly correlated with any of the other variables (*p* > 0.1). The majority of women (*n* = 283; 67.5%) held the view that children who consume mother’s milk are more likely to possess superior immunity than those who are fed with formula milk. The correlation was found to be significant with age (X^2^ = 14.047, df = 3, *p*-value = 0.002841) and education (X^2^ = 9.6491, df = 2, *p*-value = 0.00803).

The respondents most frequently selected the following advantages of breastfeeding:Closeness with a child (357 respondents, 17.7%);Building a relationship (314 respondents, 15.6%);Easy and quick access to food (317 respondents, 15.7%);Lower costs (220 respondents, 10.9%);Increasing the child’s immunity (348 respondents, 17.2%);Child’s sense of security (280 respondents, 13.9%);Reducing the risk of mother developing breast cancer (178 respondents, 8.8%).

### 3.3. Barriers towards Breastfeeding

A substantial majority of women (95.5%, *n* = 400) indicated that in case of difficulties with lactation, a certified lactation consultant should be accessible to assist a young mother. It is a matter of concern that as many as 61% (*n* = 259) of mothers reported that they did not receive sufficient help in the hospital in terms of learning how to breastfeed. The majority (67%, *n* = 280) also did not utilise the services of a lactation consultant, which was not correlated with any of the variables under the study. Similarly, the majority (65%, *n* = 271) answered that there was no lactation consultant in their place of residence. This was found to correlate with the age of the mother: X^2^ = 15.37, df = 6, *p*-value = 0.01757 (Table 3).

We also asked, “Did having a caesarean section or having a premature birth affect how you feed your baby?”. Of the 201 women who answered this question, 58 answered, “Yes, I had problems, but with help I managed to overcome them and was able to breastfeed”, and 33 answered, “Yes, unfortunately I had to bottle-feed”.

Next, Table 4 presents respondents’ answers to the question “Does breastfeeding make it difficult to return to work?”. Only 29.5% of mothers said definitely *YES*, and the differences in the answers varied depending on the place of residence.

In the present study, 181 women (43.2%) returned to work without ceasing breastfeeding. The choice of breastfeeding method was found to be dependent on the return to work (X^2^ = 42.683, df = 12, *p*-value = 2.554 × 10^−5^). This question concerned the return to work during the infant’s first year of life, defined as before the child reaches one year of age. The remaining women did not resume their professional activities at that juncture. Among the respondents who combined work with breastfeeding, 48.6% were mothers aged 30 < 40, and 73.5% had obtained a higher education qualification. No women who gave birth to a child after the age of 40 returned to work while breastfeeding (Table 5).

In our group of mothers, 42% (*n* = 176) of respondents indicated that they experienced feelings of embarrassment when breastfeeding in a public place, with some even choosing to refrain from feeding their child in such an environment (no correlation between variables, *p* > 0.05).

The most frequently indicated barrier to breastfeeding in a public place was the lack of a suitable location where a woman would feel comfortable, calm, and intimate. This view was held by 73% of the women surveyed (see Table 6). The opinions expressed on this subject did not differ from any of the variables analysed (*p* > 0.05).

## 4. Discussion

During the initial decades of the 20th century, the prevailing view was that artificial feeding was as effective as, or even more so than, natural feeding. The pioneering research conducted by Gerard, who in 1974 demonstrated the presence of immune complexes in human milk, proved to be a significant contribution to the field. From that point onwards, breastfeeding was regarded as the crucial factor in providing the infant’s immune system with protection during the initial months of life. This method of feeding children is currently recommended by numerous institutions, including the United Nations (UN), the United Nations Children’s Fund (UNICEF), the World Health Organization (WHO), the European Commission’s Directorate-General for Public Health and Risk Assessment (ECDPHRA), as well as scientific societies. The American Academy of Pediatrics (AAP), the European Society for Pediatric Gastroenterology, Hepatology and Nutrition (ESPGHAN) [6,7,33], the Polish Society for Pediatrics, Gastroenterology, Hepatology and Nutrition [3], and the Polish Academy of Sciences [4].

Nowadays, women are aware of the benefits of breastfeeding, as well as the option of using formula milk, which can be best suited to the child’s needs [15,27].

In order to promote the significant advantages associated with natural feeding for both children and mothers, the Ministry of Health in Poland initiated a campaign entitled “Milk has power” as part of the National Health Programme [34]. Conversely, World Breastfeeding Week represents one of the largest joint WHO/UNICEF campaigns, which has been celebrated annually in approximately 120 countries since 1991. The campaign is held from 1 to 7 August, although some countries organise campaigns in May [in Poland from 26 May to 1 June], October, or November [19,35,36,37]. In Poland, this initiative is also endorsed and promoted by, among others, the Association of Breastfeeding Dissemination Committee [38] and the Association of Malyssak [39]. Despite the implementation of numerous initiatives and campaigns that promote and encourage breastfeeding, the actual rates of breastfeeding remain unacceptably low [39,40].

The majority of pregnant women express a strong intention to breastfeed. In the present study, the majority of women (83.1%) indicated that they had planned to breastfeed before the birth of their child. Conversely, only a 64.4% actual breastfeeding rate was observed among the surveyed respondents following the birth of their child. In European conditions, 80–90% of pregnant women declare their desire to feed naturally in the first six months of their child’s life [27].

Despite the statements of breastfeeding mothers, there is a discrepancy between their intentions and their actual practices, a finding that is also reported by other researchers [37].

An analysis of the available literature on breastfeeding experiences, factors shaping these experiences, and attitudes towards breastfeeding allows scientists and public health professionals to examine how different personal and institutional factors influence breastfeeding decisions made by mothers. Such knowledge is essential for the identification of strategies to enhance breastfeeding rates and to facilitate a more positive and meaningful experience of breastfeeding for new mothers [39,40].

Globally, women face a multitude of obstacles that impede their capacity and willingness to breastfeed for the recommended duration or as required by the mother and infant. The practice of breastfeeding is influenced by a variety of socio-economic, cultural, and individual factors, as well as the presence or absence of public policies that promote, protect, and support breastfeeding [1,10,16,23,38]. Additionally, psychological, family-related, and public-facing factors (such as hospitals, medical personnel, workplaces, and public spaces like parks and restaurants) contribute to the complex landscape of breastfeeding experiences. Furthermore, combinations of these different environments play a pivotal role in promoting the mother’s desire to breastfeed [9,18,37].

The process of breastfeeding is a complex one that requires the acquisition of a range of skills and the development of self-confidence, as well as adequate support and assistance, especially in the first hours after giving birth in the hospital [39,40].

In the study conducted by Cierpka et al. [41], 80% of the mothers surveyed indicated that the medical staff in the ward were a source of knowledge about breastfeeding, while 86.14% cited them as a source of information about proper breastfeeding techniques. In contrast, the results of our own research indicated that the respondents did not receive adequate lactation assistance in the hospital, with 61% reporting this to be the case.

Furthermore, the absence of guidance from medical professionals regarding breastfeeding techniques contributes to women’s hesitancy and isolation, as the initial stages can be particularly challenging. Among the surveyed women, only 35% reported convenient access to breastfeeding counsel, while 95.5% asserted that it would be beneficial for mothers to breastfeed for an extended period.

In Poland, the inaugural CDL certificates were awarded to 18 consultants in 2007 in Warsaw during a conference held in celebration of World Breastfeeding Week. The number of Certified Breastfeeding Counsellors currently stands at 997 [42]. As previously stated, Poland experiences over 300,000 births annually (305,000 in 2022). This equates to approximately 306 young mothers under the care of a single counsellor. This is clearly insufficient. A further issue is that there is a lack of awareness of the availability of certified counsellors and a dearth of support from other health professionals [24,39]. At present, all maternity hospitals are required to protect, promote, and support breastfeeding [24]. The designation “Child-Friendly Hospital” is an international title that attests to the implementation of breastfeeding procedures in the care of mothers and children [38]. The most recent data from Poland indicate that only 23 of the 96 hospitals in the country are entitled to use the designation of “Child-Friendly Hospital” [38]. Other authors have highlighted the role of financial difficulties in this context [19]. Consequently, the WHO has emphasised the necessity of increasing funding for breastfeeding programmes with the aim of enhancing the proportion of mothers who breastfeed [37]. It is also noteworthy that, according to Theurich et al. [20], national breastfeeding promotion, protection, and support plans were implemented in only six of eleven countries included in the study.

The rising proportion of births by caesarean section in the Polish population also appears to exert an influence on the decision to practise mixed feeding [27]. This has led to an increase in the use of mixed and/or milk formulas, as mother-to-baby contact is delayed [43]. A correlation has been identified between caesarean section and an increased probability of delayed breastfeeding in a variety of countries [44].

In Poland, the proportion of caesarean sections is notably high, reaching 48% [27] (comparable to the prevalence observed in our group). In the study, nearly 47% of women reported having a caesarean section, which made breastfeeding difficult for some of them after giving birth, and they had to bottle-feed their babies. Perhaps the mothers did not receive sufficient professional support. Following a caesarean section, the infant should be breastfed as much as possible. A caesarean section is an acute event, occurring suddenly, which results in inadequate time for the hormones to reach a level sufficient for the production of breast milk. Nevertheless, this does not constitute an obstacle to effective breastfeeding. The fundamental principle that should be observed following the successful conclusion of a birth is the immediate attachment of the newborn to the mother. It should be noted that hospital practices may vary. It is beneficial for both the infant and the mother to have rapid skin-to-skin contact [45]. Given the mother’s greater indisposition (anaesthesia, sutures, pain, reduced mobility), it is the responsibility of the hospital staff to assist her in initiating feeding and surviving the challenging initial days following the procedure. Meanwhile, as Perrella et al. report in their research, pain and limited mobility, but also conflict and rush to care for the mother of medical personnel, i.e., lack of adequate help and support, had a negative impact on breastfeeding after caesarean section [45]. Other studies also highlight the importance of qualified midwives in rapidly adjusting the baby to the breast after caesarean section and improving breastfeeding rates [44].

It is therefore crucial to promote exclusive breastfeeding among lactating mothers by fostering a positive perception of breastfeeding [46,47]. In our study, the most frequently cited benefits of breastfeeding were the closeness of the mother to the child (17.7%), the improvement of the child’s immunity (17.2%), and the ease and speed with which food can be accessed (15.7%). The knowledge and favourable attitude of the nursing mother herself influence her decisions to breastfeed and to continue despite the difficulties encountered. The majority of women view breastfeeding as a positive experience and believe that it has numerous benefits [40]. The primary benefits of breastfeeding were perceived to be the improvement of the emotional bond between mother and child, the strengthening of the child’s immune system, and the enhancement of the mother’s self-esteem. The convenience of breastfeeding (e.g., low cost, availability) and the positive effect on postpartum weight loss were identified as additional factors that contribute to a positive perception of breastfeeding among mothers [40].

However, as the authors of the study [1,10] highlight, inadequate support from healthcare professionals represents a significant obstacle to breastfeeding. This is due to a lack of knowledge and expertise in advising and promoting the use of breast milk substitutes among mothers, a lack of clinical priority for breastfeeding, and overcrowded workloads. While breastfeeding is widely regarded as a healthy and appropriate form of infant nutrition, it is acknowledged that for some mothers, it can be a challenging and demanding process. A sense of guilt is frequently reported by non-breastfeeding mothers [37]. The findings of our study indicated that the majority of mothers expressed discontent with the necessity to alter their feeding practices. It is more common for mothers to experience psychological stress following a caesarean delivery, which is associated with suboptimal breastfeeding outcomes. Moreover, difficulties in breastfeeding in the early postpartum period are more prevalent in cases of unplanned caesarean delivery [45]. It is therefore crucial to adopt a personalised approach when supporting each woman, offering her the necessary psycho-emotional assistance and guidance. This should include a positive attitude and the support of her partner/spouse, family, and medical professionals [48].

A further factor influencing the duration of breastfeeding is the mother’s aspiration to return to work and the employer’s provision of favourable conditions for natural breastfeeding. In Poland, maternity leave lasts for a period of 20 weeks. The recommendations for breastfeeding indicate that the infant should be exclusively breastfed for up to six months, with the introduction of solid foods at the appropriate time [3,4].

Individuals who attempt to integrate childcare responsibilities with their professional obligations tend to possess a higher level of education and are currently in the optimal phase of their professional development. A period of 30–40 years is typically characterised by stability, yet the necessity for further development remains. Mothers with only one child opted for mixed feeding, whereas those with multiple children chose either to remain at home and breastfeed or to switch to modified milk.

The ease with which women returned to work while breastfeeding differed according to their place of residence. It is interesting to note that the place of residence did not have an impact on the actual return to work. Further research is required to identify the additional factors that facilitate the balance between work and household responsibilities for women.

A meta-analysis conducted by Dutheil et al. suggests that returning to work may be one of the reasons women shorten or stop breastfeeding [49]. In our study, women also indicated that returning to work may make breastfeeding difficult, but these can be reconciled with partner support or flexibility in the work schedule. The findings of Yu et al. highlight the key role of lactation rooms in creating an enabling environment for working mothers to continue breastfeeding after returning to work [28]. To facilitate the continuation of natural breastfeeding, mothers are also offered part-time work, paid breastfeeding breaks, and employer and co-worker support [49].

In our research, a significant proportion of the women surveyed (73%) identified a lack of appropriate facilities for breastfeeding in public spaces as a key issue. Additionally, 62% of the respondents reported experiencing a lack of acceptance when breastfeeding in public. It is frequently the case that public breastfeeding is met with disapproval despite the fact that it should be regarded as a natural and acceptable activity.

The act of breastfeeding a newborn, infant, or young child is done “on demand”, which means that it can occur at any time and in any location when the child expresses a need for nourishment. This indicates that the necessity to feed the infant may arise during social gatherings, professional settings, travel, outdoor activities, or walks [5,38].

Among the numerous reasons for the premature cessation of breastfeeding is women’s reluctance to do so in public settings due to a lack of social acceptance for breastfeeding women, coupled with fear and social embarrassment associated with breastfeeding in the presence of others [47].

It is therefore essential that, in any public space where a breastfeeding mother may be present, conditions are provided for the unfettered, safe, and hygienic breastfeeding of the baby [38] in order to ensure comfort and discretion during breastfeeding. This can help to minimise the discomfort of the mother, baby, and others around them [5].

In a survey conducted by Grzyb et al. [5], over three-quarters of the female respondents admitted to breastfeeding their children outside the home. However, many of them expressed feelings of shame when breastfeeding in public places. The respondents repeatedly indicated that they do not feel at ease when their child is breastfed outside the home. One in ten individuals has directly experienced disparaging comments or even criticism while breastfeeding in a public setting. As the authors observe, there is no reason to breastfeed, nor is there social acceptance for the sight of a breastfeeding mother [5]. In our survey, 73% of women also indicated a lack of a suitable place to feel free, calm, and intimate and a lack of social acceptance (62.3% of respondents indicated). The avoidance of breastfeeding in public by women can be attributed to a combination of factors, including feelings of embarrassment and fear of discomfort experienced by others, as well as their negative assessment of the situation. Furthermore, the lack of epidemiological safety and discomfort for the child are emphasised.

Other researchers have also highlighted the discomfort experienced by women who are breastfeeding in public spaces in other countries [50,51]. In a study by Gallagher et al., a comparison was made between four European countries (Sweden, Spain, Scotland, and Italy). It was observed that mothers who held negative attitudes towards breastfeeding in public places were less likely to breastfeed in public and that those who had never breastfed in public stopped doing so earlier than those with positive attitudes. The identified causes of the barriers to breastfeeding, which have been observed to result in premature cessation, include limited freedom and social isolation. Furthermore, the introduction of alternative breastfeeding methods has been identified as a potential risk factor for premature cessation of breastfeeding [43,47].

Given the well-documented benefits of breastfeeding, it is crucial to provide women with the necessary support and education to make informed decisions about their infant feeding options. Additionally, it is essential to expand public knowledge about this subject and normalise the practice of breastfeeding. The choice of whether to feed the infant with breast milk or modified milk should be made by the mother on an individual basis, taking into account her mental and physical wellbeing, as well as that of the infant [48].

In order to advance the promotion, protection, and support of breastfeeding, it is essential to implement updates to the parental social protection policy, with the aim of creating and maintaining adequate public facilities for breastfeeding. These facilities should be available not only in the workplace but also in healthcare facilities, communities, and public places. In addition, it is recommended that governments and organisations promoting breastfeeding, health systems, workplaces, communities, and parents be involved in order to enable them to play a greater role in empowering families and maintaining breastfeeding-friendly environments [38]. Furthermore, priority should be given to developing strategies to reduce challenges, increase breastfeeding confidence, and build a society where breastfeeding is a cultural norm [5,47,52].

Our study has some limitations. As is the case with other descriptive studies, the study group is not representative. To avoid this, we tried to obtain responses from mothers from all over Poland by selecting two nationwide groups on social media. The advantage is that the structure of the participants is similar to that observed in other studies on related topics, and to the structure of the Polish female population (and ways of pregnancy termination), it may be reasonably assumed that the respondents were drawn from across the country. In addition, we excluded groups thematically related to breastfeeding because they may bring together women who particularly want to breastfeed (which may be a confounding factor in terms of, for example, the perception of barriers). It is therefore important to approach the question of whether the conclusions drawn from the study correspond to the opinions of the entire population with great caution [53]. However, our conclusions may prove useful for lactation educators, for those responsible for the organisation of mother and child care, or for architects designing spaces adapted to the needs of young mothers.

## 5. Conclusions

Our study shows that breastfeeding in Poland is at an unsatisfactory level. Mothers, despite noticing the many benefits of breastfeeding, encounter barriers that can make it difficult. A common problem reported by respondents was insufficient care and support from medical personnel, including support from a lactation consultant. Another problem was returning to work, but through family support and flexible working hours, these difficulties can be overcome. Breastfeeding in public places was also a barrier—feeling uncomfortable because there were no appropriate places adapted for natural breastfeeding. Therefore, it is very important to pay more attention to properly prepared, educated medical personnel, including lactation consultants, so that they can support mothers in their decision and physically help with breastfeeding in the first hours after birth and to create places in public spaces that are more adapted to breastfeeding (lactation rooms) to make it easier for mothers to breastfeed. Further research on a larger group of breastfeeding women in Poland would be advisable.

## Figures and Tables

**Table 1 healthcare-12-01744-t001:** Socio-demographic characteristics of the studied population.

Variable	Characteristics	Number	Percentage	95%Cl
Age (years)	18 < 25	43	10.26	0.08–0.13
	25 < 30	110	26.25	0.22–0.31
	30 < 40	206	49.16	0.44–0.54
	40 and above	60	14.32	0.11–0.18
Age of birth of first child (years)	18 < 25	118	28.16	0.24–0.33
	25 < 30	174	41.53	0.37–0.46
	30 < 40	125	29.83	0.26–0.34
	40 and above	2	0.48	0.00–0.01
Number of children	1	170	40.57	0.36–0.45
	2	177	42.24	0.38–0.47
	3	63	15.04	0.12–0.19
	4 and more	9	2.15	0.01–0.04
Education	Lower secondary and vocational	20	4.77	0.03–0.07
	Secondary general	95	22.67	0.19–0.27
	University	304	72.55	0.68–0.77
Place of residence	City >100,000 inhabitants	200	47.73	0.43–0.53
	Rural and city <100,000 inhabitants	219	52.27	0.47–0.57
Method of termination of pregnancy *	Naturally (full-term pregnancy)	200	49.88	0.45–0.55
	Naturally (premature birth)	13	3.24	0.02–0.05
	Caesarean section (full-term pregnancy)	147	36.66	0.32–0.41
	Caesarean section (necessary earlier termination of pregnancy)	41	10.22	0.08–0.13

* 401 women answered this question.

**Table 2 healthcare-12-01744-t002:** Comparison of how respondents wanted to feed their infants and how they did.

Type of Feeding	How Did Mother Want to Feed?		How Did Mother Actually Feed?	
	Number (%)	95%CI	Number (%)	95%CI
Breastfeeding	348 (83.1)	0.79–0.86	270 (64.4)	0.60–0.68
Formula milk	23 (5.5)	0.04–0.07	30 (7.2)	0.05–0.09
Mixed method	48 (11.4)	0.09–0.15	119 (28.4)	0.24–0.32

**Table 3 healthcare-12-01744-t003:** Selected barriers to breastfeeding vs. place of residence.

Questions	Variable: Place of Residence	No Number (%)	Yes Number (%)
Did you receive enough help in the hospital to learn how to breastfeed after your baby was born?	City >100,000 inhabitants	126 (63%)	74 (37%)
	Rural and city <100,000 inhabitants	133 (61%))	86 (39%)
After you left the hospital, did you seek help from a lactation consultant?	City >100,000 inhabitants	126 (63%)	74 (37%)
	Rural and city <100,000 inhabitants	154 (70%)	65 (30%)
Is there easy access to lactation advice where you live?	City >100,000 inhabitants	123 (61.5%)	77 (38.5%)
	Rural and city <100,000 inhabitants	148 (68%)	71 (32%)

**Table 4 healthcare-12-01744-t004:** Answers to the question “Does breastfeeding make it difficult to return to work?”.

Variable	No, It Does Not Hinder	Yes, Rather Yes, But It Can Be Reconciled Somehow (e.g., Partner Support, Change in Working Hours)	Yes, It Definitely Makes It More Difficult	*p*-Value
Place of residence				0.03594
City >100,000 inhabitants	45 (22.5)	106 (53%)	49 (24.5%)	
Rural and city <100,000 inhabitants	32 (14%)	113 (51%)	74 (34%)	
Number of children				0.05463
1	21 (12%)	99 (58%)	50 (29%)	
2	42 (24%)	79 (44%)	56 (32%)	
3	11 (17%)	36 (57%)	16 (25%)	
4 and more	3 (0.3) *	5 (0.5) *	1 (0.1) *	

* Due to the number of cases, the frequency is given in fractions.

**Table 5 healthcare-12-01744-t005:** Characteristics of mothers who returned to work without stopping breastfeeding (between 6 months and 1 year of the infant’s life).

Variable	Characteristics	Number (%)	95%CI	*p*-Value
Age (years)	18 < 25	18 (9.94)	0.06–0.14	0.47
	25 < 30	47 (25.97)	0.20–0.32	
	30 < 40	88 (48.62)	0.41–0.55	
	40 and above	28 (15.47)	0.11–0.21	
Age of birth of first child (years)	18 < 25	49 (27.07)	0.21–0.34	0.32
	25 < 30	75 (41.44)	0.34–0.49	
	30 < 40	57 (31.49)	0.25–0.38	
	40 and above	0	0	
Number of children	1	61 (33.70)	0.27–0.40	0.00032
	2	96 (53.04)	0.46–0.60	
	3	20 (11.05)	0.07–0.16	
	4 and more	4 (2.21)	0.01–0.05	
Education	Lower secondary and vocational	5 (2.76)	0.01–0.06	0.02
	Secondary general	35 (19.34)	0.14–0.25	
	University	141 (77.90)	0.71–0.83	
Place of residence	City >100,000 inhabitants	91 (50.28)	0.43–0.57	0.11
	Rural and city <100,000 inhabitants	90 (49.72)	0.42–0.57	

**Table 6 healthcare-12-01744-t006:** Women’s answers to the question “What, in your opinion, can be a barrier to breastfeeding a child in a public place?”.

Barriers to Breastfeeding in Public Place	Number	Percentage
Lack of acceptance by society	261	62.29
Lack of intimate places where mothers can breastfeed their babies	305	72.79
Fear of the gaze of others	208	49.64
Being ashamed	177	42.24
The feeling of nakedness	163	38.90

## Data Availability

The raw data supporting the results and the conclusions of this article will be made available by the corresponding author upon request.

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
