# Peer review of "Attitudes and Barriers of Polish Women towards Breastfeeding—Descriptive Cross-Sectional On-Line Survey"

_healthcare, 2024, doi:10.3390/healthcare12171744_

Round 1
Reviewer 1 Report
Comments and Suggestions for Authors
Dear authors,
The manuscript's topic is very current and significant in terms of a very challenging issue in public health. With the study, you have tried to make a new contribution and provide additional empirical data on women's attitudes and barriers to breastfeeding in Poland. However, significant improvements are necessary for the manuscript to provide new valid empirical data to complement the highly respectable existing literature data on breastfeeding in Poland.
I would like to make suggestions for improving the manuscript:
- Abstract: In line 11, you mention young mothers, while in the manuscript, they are not the exclusive target group. Consider reformulating the sentence. The results are dominated by sociodemographic and general data and not specific data on attitudes and barriers related to breastfeeding. Please correct it. In conclusion, various risk factors are mentioned, and the study did not examine risk factors for breastfeeding. Please correct it.
- Keywords: you chose Opinions as a keyword and not Attitudes. What determined your decision?
- Introduction: The advantages of breastfeeding for both mother and child are listed, with special reference to the nutritional needs of the child and its positive impact on the child's development and health. In order for a more complete introduction, it is necessary to specify specific data, e.g. enter the data in line 80 with the year of determining the rate. Also, literary empirical data from studies of different designs indicates the rates of breastfeeding and rates of exclusive breastfeeding, as well as other different aspects of breastfeeding in Poland. You mention some of them in the discussion. They should be listed/transferred in this section. Considering that for Poland, a number of data have been collected in well-designed studies, mostly using validated instruments, it would be desirable if you could state in a few sentences the reasons for your study.
Technical suggestions: Line 34 has a full stop/period after the word necessary. Is that a typo or missing part of the sentence? Please correct it. Also, the beginning of the sentence in line 54 requires a correction.
- Materials and Methods: In the study's design, it is necessary to clearly state and explain the criteria for analyzing the posts and topics, who performed the analysis, and clearly present the criteria for including mothers and what these criteria are based on.
What is the basis of your point of view stated in lines 132-133 (international, Poland, institutional of your institution)? Please support it with a reference.
In line 137, after the existing literature, indicate which literature it specifically refers to. The procedure for creating the questionnaire needs to be clearly described. It is necessary to describe all the steps more precisely. It is unclear whether any of the students involved in the "process" have become "mothers" or whether they have had experience with breastfeeding. Indeed, the statement that the process has provided an assessment of the reliability and validity of the survey requires that the parameters that confirm this be stated in full.
Since the survey contains three different sections, each section should be described in a separate paragraph for a better understanding of the methodology and the results of the study. It would also be useful to attach the entire survey in English to the manuscript.
In line 146 it would be useful to specify the range of answers.
The statements in lines 153-154 about the caesarean section and the explanations related to breastfeeding are not entirely clear for this part of the manuscript. What is their contribution to this part?
- Results: Tables 1, 3, 4, and 5 should be edited technically; the merge-cells option for variables is not applied. Fix it. Specific characteristics are found in several rows, for example 25 years (14-25/25-30 and 30-40/40 and over), which is unacceptable because the question arises in which group are mothers who are 25, 30 or 40 years old? the same for the next variable as for place of residence. In the name of tables 1, 2, 4, 5 and 6, it is unnecessary to mention the total number of respondents whose data are in the table.
It would be very useful to present the data listed in lines 181-185 in a table, both as a whole and according to the respondents' sociodemographic characteristics. It would also be essential to analyze the data from Table 6 about these characteristics of the respondents.
Technical suggestions: In the data representation, the sign for the test (X2 / X-squared) must be shown uniformly.
- Discussion: The authors draw attention to many important questions in the area of ​​breastfeeding. However, according to the previous suggestions, some data can be transferred to the Introduction. Also, they did not link many of the literary data they mentioned and point to critical issues related to breastfeeding with the results of their study (lines 336-341; 369-394; 456-468). References are missing for the statements in lines 302-303, 313-314, 435-439 and 492-497.
- Limitations: The authors identify those aspects that generated limitations; however, it would be important to explain the actions they took to minimize them.
- Conclusion: Very general; please be more specific; especially in relation to the goals set. Risk factors are also mentioned, but this was not the aim of the study, nor were the data analyzed by factor analysis.
I hope you find my comments helpful.
Author Response
Dear Reviewer #1,
We are very grateful for your valuable suggestions, which allowed us to improve the manuscript. Below we present our responses point by point. We have marked the changes in the manuscript in red.
1a)- Abstract: In line 11, you mention young mothers, while in the manuscript, they are not the exclusive target group. Consider reformulating the sentence. – Response: yes, You are right, it was inaccurate; when we wrote "young mothers", as we also wrote in the methodology, we meant mothers who gave birth in the last 5 years (not mothers who were young by age); but in the Abstract it can be misleading and is really unnecessary; we deleted this fragment.
1b) The results are dominated by sociodemographic and general data and not specific data on attitudes and barriers related to breastfeeding. Please correct it. – Response: Dear Reviewer. Thank you for this question. To explain only two lines (16 and 17) refer to sociodemographic data; the remaining results in the abstract concern attitudes (ex. „83.1% of mothers planned to breastfeed, but not all of them were able to do so for various reasons; One third of them felt sorry for themselves that they had to change their feeding method” (lines 18-19), and also barriers (ex. „mothers did not receive sufficient help in the hospital in terms of learning how to breastfeed (61%), did not use the help of a lactation consultant (67%), and answered that there was no lactation consultant in their place of residence (65%)”, but also barriers related to returning to work and breastfeeding in public places.
1c) In conclusion, various risk factors are mentioned, and the study did not examine risk factors for breastfeeding. Please correct it. - Response: yes, we agree, we have corrected it according to the reviewer's suggestion.
2)- Keywords: you chose Opinions as a keyword and not Attitudes. What determined your decision? - Response: Thank you for this attention. Opinion is someone's opinion about someone or something. Attitude is related to action. Here we were actually more interested in how women behave and the word "Attitudes" would be more appropriate (as it is in the title of the manuscript).
3)- Introduction: The advantages of breastfeeding for both mother and child are listed, with special reference to the nutritional needs of the child and its positive impact on the child's development and health. In order for a more complete introduction, it is necessary to specify specific data, e.g. enter the data in line 80 with the year of determining the rate. Also, literary empirical data from studies of different designs indicates the rates of breastfeeding and rates of exclusive breastfeeding, as well as other different aspects of breastfeeding in Poland. You mention some of them in the discussion. They should be listed/transferred in this section. Considering that for Poland, a number of data have been collected in well-designed studies, mostly using validated instruments, it would be desirable if you could state in a few sentences the reasons for your study. – Response: Thank you for this attention - we have completed the missing data on infant feeding rates in the world, in Europe and in Poland. We have also moved, as suggested by the Reviewer, some issues related to breastfeeding from discussion to introduction. We provided the justification for the studies undertaken - this was a very important addition and explanation, which the Reviewer drew attention to - for which we thank you very much.
Due to the transfer of some information and the addition of other important data – numbering in the literature has been changed.
4) Technical suggestions: Line 34 has a full stop/period after the word necessary. Is that a typo or missing part of the sentence? Please correct it. Also, the beginning of the sentence in line 54 requires a correction. - Response: Thank you - we've removed these issues – an editorial bug has crept in.
5a)- Materials and Methods: In the study's design, it is necessary to clearly state and explain the criteria for analyzing the posts and topics, who performed the analysis, and clearly present the criteria for including mothers and what these criteria are based on. - Response: We have corrected this fragment. We have added keywords based on which we selected groups in which the link to the survey was placed. Among all the groups, only these two had a nationwide reach, we excluded the remaining groups with a local reach (we have added this note in the manuscript). Among the keywords, we deliberately avoided the phrase "breastfeeding" so that the selected group was as neutral as possible (there is also a note about this in the manuscript).
5b) What is the basis of your point of view stated in lines 132-133 (international, Poland, institutional of your institution)? Please support it with a reference. - Response: We have supplemented with references; This was our mistake (typo) - because the reference was included in the earlier list of references, but unfortunately it was missing in the text; We apologize.
5c) In line 137, after the existing literature, indicate which literature it specifically refers to. The procedure for creating the questionnaire needs to be clearly described. It is necessary to describe all the steps more precisely. It is unclear whether any of the students involved in the "process" have become "mothers" or whether they have had experience with breastfeeding. Indeed, the statement that the process has provided an assessment of the reliability and validity of the survey requires that the parameters that confirm this be stated in full. - Response: Thank you for this comment. When creating the survey, we used the literature included in this publication, where we cite/compare our results to the results of other authors.
In order to clarify the doubt regarding the student's participation - she did not become a mother, but she conducted the study on a smaller group of mothers as part of her master's thesis. Thanks to this, we checked whether the questions were understandable, and also changed several answer choices. We believe that her contribution helped us in our study, which is why we thanked the student for her help in the "Acknowledgements" section.
5d) Since the survey contains three different sections, each section should be described in a separate paragraph for a better understanding of the methodology and the results of the study. It would also be useful to attach the entire survey in English to the manuscript. - Response: We have separated the mentioned paragraphs and included the questionnaire translated into English in the supplementary materials.
5e) In line 146 it would be useful to specify the range of answers. - Response: We have supplemented it according to the reviewer's suggestion.
5f) The statements in lines 153-154 about the caesarean section and the explanations related to breastfeeding are not entirely clear for this part of the manuscript. What is their contribution to this part? - Response: This issue arose here because we treated cesarean section as a possible barrier to breastfeeding. However, we actually agree with the Reviewer that it would be better to include this information in the sociodemographic data, and that is what we did.
6a)- Results: Tables 1, 3, 4, and 5 should be edited technically; the merge-cells option for variables is not applied. Fix it. Specific characteristics are found in several rows, for example 25 years (14-25/25-30 and 30-40/40 and over), which is unacceptable because the question arises in which group are mothers who are 25, 30 or 40 years old? the same for the next variable as for place of residence. In the name of tables 1, 2, 4, 5 and 6, it is unnecessary to mention the total number of respondents whose data are in the table. - Response: Yes, we agree with the Reviewer. We will merge cells in the manuscript editing process. The ranks were well chosen in the questionnaire. We have corrected this aspect in the manuscript, in accordance with the reviewer's comments. Due to the size of the groups, we have aggregated the answers for rural and city < 100.000 inhabitants for calculations. We have deleted the numbers of respondents in the table titles.
6b) It would be very useful to present the data listed in lines 181-185 in a table, both as a whole and according to the respondents' sociodemographic characteristics. It would also be essential to analyze the data from Table 6 about these characteristics of the respondents. – Response: Dear Reviewer, We did various analyses according to various characteristics, but unfortunately no correlations emerged, therefore due to the extensiveness of the publication, we did not include the results that we did not consider interesting. When it comes to self-assessment of knowledge, this is a subjective feeling of women, and the vast majority believed that they had this knowledge (both knowledge about breastfeeding and knowledge about feeding infants). As many as 77.3% and almost 84%, respectively, believed that they had good or very good knowledge. And it seems to us that writing this down in a table according to sociodemographic characteristics would not contribute anything to the manuscript. No correlations were found here, and the self-assessment study was not the aim of this study either. In turn, when it comes to Table No. 6, it is similar. Table 6 presents the barriers most frequently mentioned by mothers, but there were no differences in the responses in relation to sociodemographic data, which we regret, because we actually thought that the barriers were perceived differently by women of different ages, or by women living in the city/village. It turned out that the barriers were "common" for many mothers, and we mentioned the lack of correlations in relation to sociodemographic characteristics in the manuscript.
6c) Technical suggestions: In the data representation, the sign for the test (X2 / X-squared) must be shown uniformly. – Response: we have corrected.
7)- Discussion: The authors draw attention to many important questions in the area of ​​breastfeeding. However, according to the previous suggestions, some data can be transferred to the Introduction. Also, they did not link many of the literary data they mentioned and point to critical issues related to breastfeeding with the results of their study (lines 336-341; 369-394; 456-468). References are missing for the statements in lines 302-303, 313-314, 435-439 and 492-497. – Response: According to the Reviewer’s suggestion we have moved some information about breastfeeding to the introduction, and some issues have been removed. We have removed information from line 337-341 – we have added other information; 369-379 – some content from this part has been moved to the introduction; lines 380-393 – we have added specific information. Messages from line 393-394 and 448-451- and 456-464– we have removed. In the given lines: 302-303, 313-314, 435-439 and 492-497 - we have added information from the literature.
8)- Limitations: The authors identify those aspects that generated limitations; however, it would be important to explain the actions they took to minimize them. - Response: We have corrected this fragment in the manuscript.
9)- Conclusion: Very general; please be more specific; especially in relation to the goals set. Risk factors are also mentioned, but this was not the aim of the study, nor were the data analyzed by factor analysis. – Response: We have changed the Conclusions section.
I hope you find my comments helpful. Yes, the reviewer's comments were very helpful. Thank you.

Reviewer 2 Report
Comments and Suggestions for Authors
This study tried to identify Attitudes and barriers of Polish women towards breastfeeding 2 – descriptive cross-sectional survey, using a cross-sectional study. The explanation for choosing this lot is incomplete and not sustained.
The topic is relevant at this Population, but the value of this lot should be explained. What is the Polish population? What is the significance of percents?
This study tries to address a specific gap in this field.
This study tried to present actual data in the Polish Population. The authors explain the importance of breastfeeding in Poland. Attitudes and barriers are identified and well presented.
The methodology is not adequate and is not well expressed. The authors presented in the article that an informed consent was obtained.
Please add as relevant material (an example of informed consent).
The questionnaire as relevant material – please present it as a blank example.
The authors compared the lots considering the age (between them), but this importance of study will increase if they will compare considering the significance with all population in Poland.
Generally speaking, the authors could improve the conclusion of article.
References are appropriate and well-chosen.
The references are carefully chosen, from impact journals. They are updated, from the last 10-12 years, in this area of medicine. They constitute a real support for the text of the article.
Tables are very easy to understand, comparing the groups of population integrated into the cross-sectional study. They include mean, standard deviation, and statistical significance. It is very simple and detailed at the same time, supporting the conclusion of the cross-sectional study, but some details should be improved.
Comments on the Quality of English Languageit is ok to understand.
Author Response
Dear Reviewer #2,
We are very grateful for your valuable suggestions, which allowed us to improve the manuscript. Below we present our responses point by point. We have marked the changes in the manuscript in red.
- This study tried to identify Attitudes and barriers of Polish women towards breastfeeding 2 – descriptive cross-sectional survey, using a cross-sectional study. The explanation for choosing this lot is incomplete and not sustained.
The topic is relevant at this Population, but the value of this lot should be explained. What is the Polish population? What is the significance of percents? – Response: The study was conducted online, the research tool was a survey questionnaire. The structure of the study group had characteristics similar to the general population (sociodemographic characteristics), but due to the selection of the group (online form) it was not a representative group, which we also explained in the methodology and in the limitations of the study. Additionally, in the Methodology section we explained how large the selected groups were in social media and which group took part in the study. We have added the title of the manuscript.
- This study tries to address a specific gap in this field.
This study tried to present actual data in the Polish Population. The authors explain the importance of breastfeeding in Poland. Attitudes and barriers are identified and well presented. Response: We thank the reviewer for this comment.
- The methodology is not adequate and is not well expressed. - Response: We have corrected this part of the manuscript.
- The authors presented in the article that an informed consent was obtained. Please add as relevant material (an example of informed consent). - Response: Thank you for your attention The survey questionnaire began with information about the purpose of the study. Then, respondents selected the appropriate response indicating their willingness to participate in the study and the appropriate age (over 18 years). In the supplementary material, we included a translated survey questionnaire, where this information is included at the beginning.
- The questionnaire as relevant material – please present it as a blank example. Response: Thank you, yes we have attached the survey questionnaire as supplementary material
- The authors compared the lots considering the age (between them), but this importance of study will increase if they will compare considering the significance with all population in Poland. - Response: Yes, that's a valuable comment. Unfortunately, our study cannot be considered a representative group, so we cannot refer our results to the entire population in Poland. This is one of the limitations of the study. We wanted to check first whether these problems exist in the opinion of women. In the next step, we will plan a study on a representative group, in which we will be able to refer to the entire population. In the Discussion section we compared the obtained results with other Polish studies.
- Generally speaking, the authors could improve the conclusion of article. - Response: We have changed (improved) the Conclusions section. Thank you.
- References are appropriate and well-chosen.
The references are carefully chosen, from impact journals. They are updated, from the last 10-12 years, in this area of medicine. They constitute a real support for the text of the article. - Response: We thank the reviewer for this comment
- Tables are very easy to understand, comparing the groups of population integrated into the cross-sectional study. They include mean, standard deviation, and statistical significance. It is very simple and detailed at the same time, supporting the conclusion of the cross-sectional study, but some details should be improved. - Response: We thank the reviewer for this comment. We have corrected the ranks of the answers so that they are not mutually exclusive.
- Comments on the Quality of English Language
it is ok to understand. - Response: We thank the reviewer for this comment

Round 2
Reviewer 1 Report
Comments and Suggestions for Authors
Dear authors,
Your corrections sufficiently improved the manuscript so that it can be published in Healthcare.
Reviewer 2 Report
Comments and Suggestions for Authors
The article is improved. The authors improved the title, methodology, and conclusions based on the first review.
This version is better and acceptable to be published.